# Predicting the failure of two-dimensional silica glasses

Francesc Font-Clos[1,4], Marco Zanchi[1,4], Stefan Hiemer[2,4], Silvia Bonfanti[1,3], Roberto Guerra[1], Michael Zaiser[2] & Stefano Zapperi[1,2,3 ✉]

Being able to predict the failure of materials based on structural information is a fundamental issue with enormous practical and industrial relevance for the monitoring of devices and components. Thanks to recent advances in deep learning, accurate failure predictions are becoming possible even for strongly disordered solids, but the sheer number of parameters used in the process renders a physical interpretation of the results impossible. Here we address this issue and use machine learning methods to predict the failure of simulated two dimensional silica glasses from their initial undeformed structure. We then exploit Gradient-weighted Class Activation Mapping (Grad-CAM) to build attention maps associated with the predictions, and we demonstrate that these maps are amenable to physical interpretation in terms of topological defects and local potential energies. We show that our predictions can be transferred to samples with different shape or size than those used in training, as well as to experimental images. Our strategy illustrates how artificial neural networks trained with numerical simulation results can provide interpretable predictions of the behavior of experimentally measured structures.

[1] Center for Complexity and Biosystems, Department of Physics, University of Milan, via Celoria 16, 20133 Milan, Italy. [2] Institute of Materials Simulation, Department of Materials Science Science and Engineering, Friedrich-Alexander-University Erlangen-Nuremberg, Dr.-Mack-Str. 77, 90762 Fürth, Germany. [3] CNR - Consiglio Nazionale delle Ricerche, Istituto di Chimica della Materia Condensata e di Tecnologie per l'Energia Via R. Cozzi 53, 20125 Milan, Italy. [4] These authors contributed equally: Francesc Font-Clos, Marco Zanchi, Stefan Hiemer. ✉email: stefano.zapperi@unimi.it

Predicting material failure based on structural information is a central challenge of materials science and engineering. The issue is particularly thorny in the case of disordered solids where the internal amorphous structure prevents a straightforward identification of isolated defects that could become the seeds for crack nucleation. Guided by molecular dynamics simulations of idealized models of glasses, it is possible to define empirical structural indicators and assess their predictive value by correlation analysis[1,2], but using them to make predictions on realistic disordered solids, such as silica glasses or polymer networks, is extremely challenging. Modern machine learning methods provide a promising alternative pathway for the systematic development of structure-based predictions[3], as has been shown in the context of molecular properties[4–6], density functional theory force fields[7–9], governing equations for dynamical systems and flow[10–12] and dislocation models for crystal plasticity[13–15]. Applications to glasses have been so far restricted to idealized models, so-called Lennard-Jones glasses, that were analyzed with support vector machines (SVM)[16–18], graph neural networks (GNN)[19] and deep learning[20,21]. Predictions based on deep learning methods are becoming increasingly accurate but they are also hard to interpret. In the context of glasses, this means that although deep learning may accurately predict when a material will fail, the structural determinants of failure often remain obscure.

Two-dimensional silica glass[22], first observed by depositing a single bilayer of $SiO_2$ molecules on a graphene substrate[23], provides perhaps the simplest example of a disordered solid. Its atomic structure and defect dynamics can be directly observed by electron microscopy[24] or atomic force microscopy[25] while molecular dynamics simulations can be used to accurately reproduce its structural features and investigate its mechanical properties[26–28].

In this work, we analyze the structure and failure behavior of two-dimensional silica glasses by machine learning methods and illustrate how accurate failure prediction can be achieved while preserving the qualitative interpretability of the results. This is achieved thanks to the use of Gradient-weighted Class Activation Mapping (Grad-CAM)[29] which allows us to visualize where the deep neural network focuses its attention when developing a prediction (see Fig. 1).

## Results

**Atomwise rupture prediction by support vector machine.** To obtain training and test sets for the machine learning approaches, we first generate a large number of realistic atomic configurations of silica bilayers with controlled in-plane disorder (see Fig. 1), which is quantified by the standard deviation $s^2$ of the ring size distribution as in previous studies[27,30] (see Methods for details). We consider three sets of data, the first two obtained in such a manner that all samples have the same level of disorder ($s^2 = 0.2$ and $s^2 = 0.3$, respectively), whereas the third contains samples of different disorder, $s^2 \in [0.25, 1]$. Each configuration is composed of $N = 3456$ atoms arranged in a square box of edge length $L \sim 85$ Å with periodic boundary conditions along the $x, y$ plane.

In our molecular dynamics simulations, atomic interactions are described by the Watanabe interatomic potential[31,32] described in the Methods section. This interatomic potential has been devised to describe $SiO_2$ surfaces and appears therefore to be appropriate for 2D silica. We have also explored different potentials commonly used to model silica, such as the Vashishta[33], Tersoff[34], and COMB10[35] potentials. Simulations show (see Supplementary Fig. 1) that the Watanabe potential yields the most aligned configurations across the two layers, in agreement with experiments[23] and ab initio calculations[36].

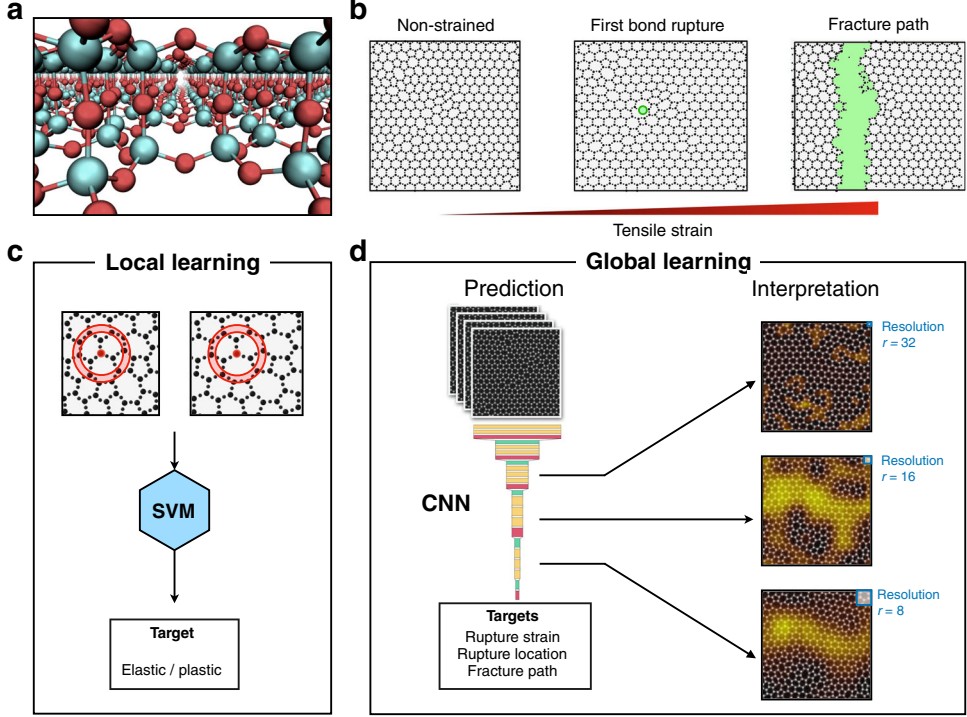

**Fig. 1 Schematic of system and algorithms. a** Perspective view of the silica glass bilayers: silica atoms are colored in cyan and oxygen ones in red. **b** Schematic representation of the fracture formation under increasing tensile strain from left to right: non-strained (left panel), first plastic event corresponding to a bond breaking (central panel) and crack path (right panel). **c** Local learning approach that uses Support Vector Machine to predict the elastic/plastic nature of individual atoms. **d** Global learning approach which uses a ResNet model to predict rupture strain, location and the full crack path. The approach also allows for the interpretation of the model decisions using attention maps (right side of panel).

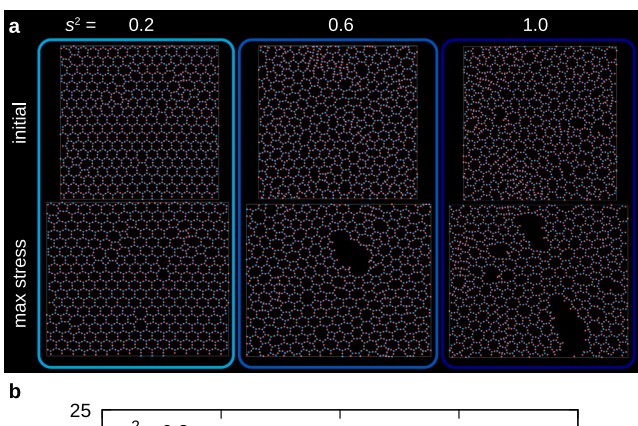

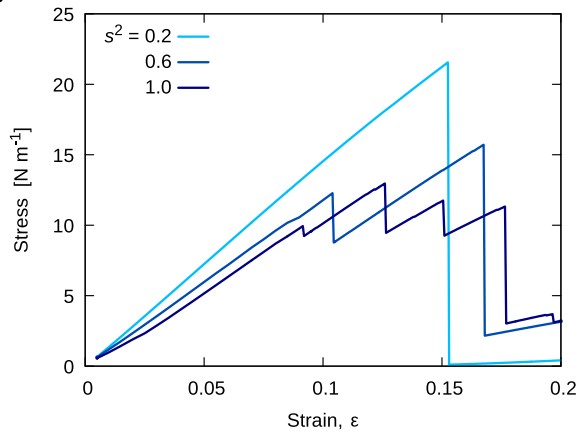

**Fig. 2 Straining of silica samples. a** Three typical configurations of silica samples with different disorder levels $s^2 = 0.2$, $s^2 = 0.6$, and $s^2 = 1.0$. We report the initial configuration and the configuration corresponding to the peak stress. **b** Stress-strain curves for the samples corresponding to panel A. The stress is measured in N/m considering a two-dimensional system of area $L_x \cdot L_y$, where $L_x$ and $L_y$ are the cell longitudinal and transversal size of the initially relaxed (unstrained) configuration. As the disorder increases, the stress-strain curves become less brittle.

Starting from the initial configuration, we progressively stretch the simulated sample along the $x$ direction by small displacement steps and subsequent relaxation using the athermal quasistatic (AQS) protocol (see Methods for details). The configuration is stretched until the system fractures. In Fig. 2a, we report typical configurations before and after straining and in Fig. 2b, the corresponding stress-strain curves for different values of disorder. We observe brittle behavior at low disorder and more ductile behavior when disorder is strong, in agreement with previous simulations[27]. In Fig. 2b, stress is measured in N/m as it is appropriate for a two-dimensional material whose thickness can not be defined precisely. From the slope of the stress-strain curves at low disorder, we estimate a Young modulus per unit thickness equal to $E/h \simeq 146$ N/m. This is in reasonable agreement with ab initio calculations which yield $E/h \simeq 132$ N/m[36].

For each configuration, we record the location of the first broken bond and the strain value at which it occurs, which we define as rupture strain. We also record the crack path, defined as the set of contiguous atoms which disconnect the sample after the final catastrophic fracture event (see Fig. 1b). These quantities will be the targets of our failure predictions. Following previous machine learning approaches to glasses[16–19], we first consider an atomic-level prediction strategy in which we evaluate atoms one by one, assessing whether their bonding state and atomic surrounding are going to change irreversibly under applied load. In the following, we say that an atom deforms irreversibly when

its bonding state changes upon straining, otherwise we define the local deformation as elastic. To take into account systematically all the system symmetries, we do not work directly with atomic positions but compute radial symmetry functions associated with the atomic positions both in the initial configuration and in the configuration obtained after the application of an affine transformation corresponding to the applied strain. In the previous approaches[16–19], no affine transformation was applied so that the algorithm would yield the same results independently on the loading condition, which is unphysical.

We apply SVM classification to the data set we constructed. The SVM is a machine learning algorithm that maps training data to points in space so that the gap between the two categories is as wide as possible. New data are then assigned to a category, depending on the side of the gap in which they fall after the same mapping is applied (see Methods). Our classification strategy turns out to be quite accurate, with a sensitivity of 99.4% and a specificity of 96%. As shown in Fig. 3a, the SVM is thus able to correctly classify almost 96% of the atoms that deform elastically (true negatives, TN), but since only less than 0.1% atoms deform irreversibly in each configuration (see Fig. 3b for two examples), the problem is heavily unbalanced and thus, despite the high specificity of the method, the rate of false positives (FP) is relatively high (more than 4%). This reflects a generic problem of fracture prediction as the fraction of atoms that participate in the actual fracture process may approach zero in the thermodynamic limit of large samples. Nevertheless, in the present case the predictor identifies a subset of particles, which contains almost all irreversibly deforming atoms, while discarding the vast majority of those deforming elastically. The local approach proves to be fairly robust against specific choice of hyper-parameters (see Supplementary Fig. 2 and the Supplementary Note 1 for more details) which just leverage a different trade-off between sensitivity and specificity (i.e., true negative and true positive predictions).

**Residual neural networks predict disorder**. Because of the nature of fracture as a multiscale problem that is equally influenced by local configurations and by system-scale interactions and correlations, methods that focus exclusively on local atomic environments have intrinsic limitations. To overcome the limitations of local methods such as SVM, we resort to global image-based predictions. To this end we generate images of a large number of particle configurations. Together with the corresponding fracture characteristics these images provide training and test sets used by a residual neural network (ResNet) for image-based recognition of structural features such as degree of disorder, and of failure-related features such as rupture location and strain, and final crack path which is encoded as an image. ResNet is a deep convolutional neural network consisting of residual blocks with short-cut connections enabling information to bypass convolutional layers to stabilize the learning process by avoiding vanishing gradients (see Methods and Supplementary Fig. 3 for further details on the ResNet we use).

We first train the ResNet to characterize the structure of different samples by identifying the disorder level associated with a given configuration taken from the variable disorder set. As shown in Fig. 3d, the ResNet is then able to accurately predict the disorder level for configurations in the test set. Next, we employ Grad-CAM to draw attention maps associated with the predictions. Grad-CAM maps can be understood as pixels-wise derivatives of a given convolutional layer with respect to the ResNet final output. The method helps us understand how the artificial network takes decisions, highlighting the regions in the

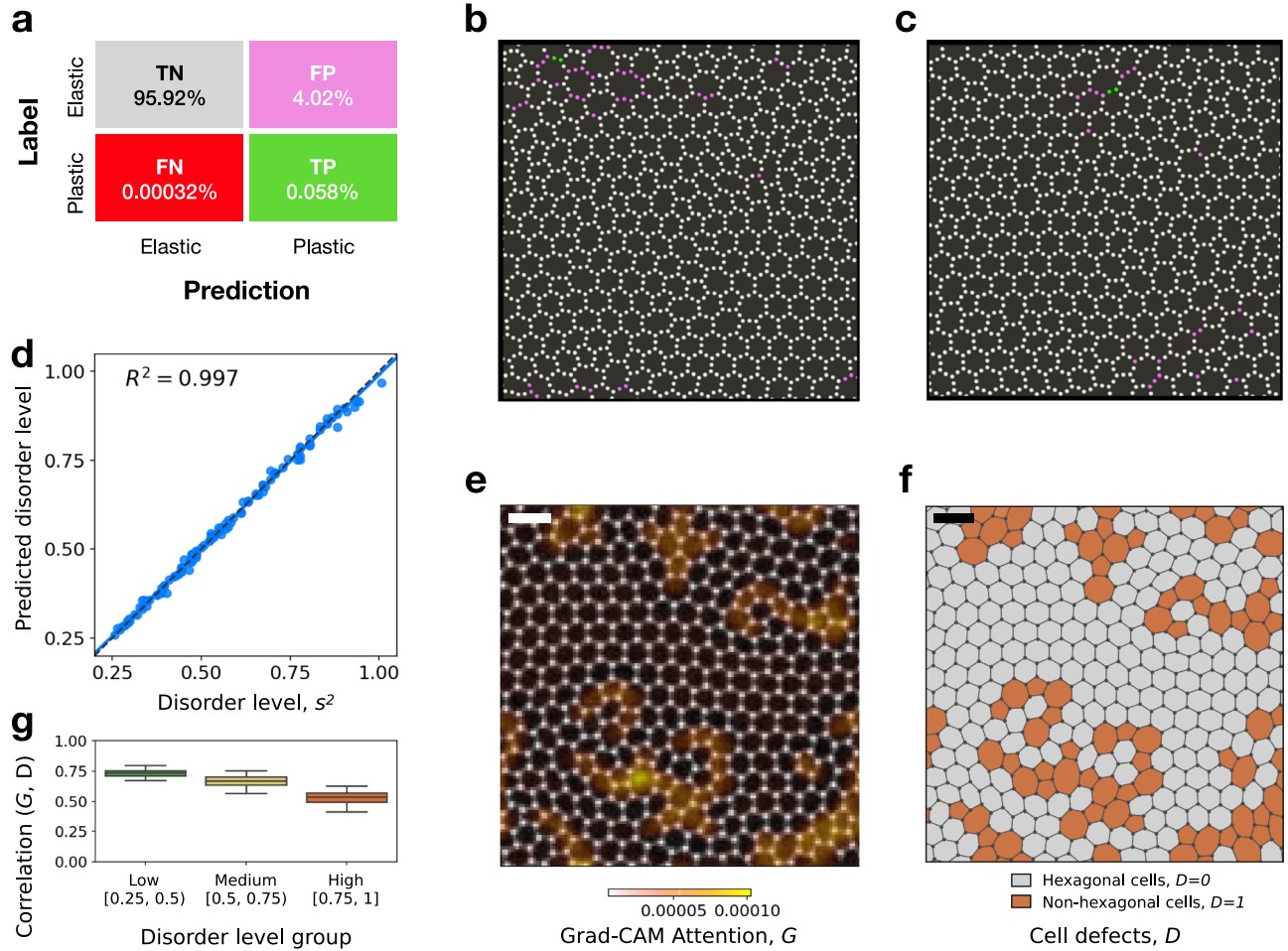

**Fig. 3 Machine learning classifies local and global disorder. a** Confusion matrix for the prediction of particles involved in the first bond fracture done by the support vector classifier. It can be clearly seen that a subset of particles is identified by the ML algorithm which includes almost all particles involved in the first bond rupture. **b, c** Prediction of an atomistic system with disorder level $s^2 = 0.2$, with coloring scheme according to the confusion matrix in a. **d** Prediction by the ResNet matches the true disorder level $s^2$ of Silica glass sheets. **e** Example of silica sample (black-and-white image) superimposed with the Grad-CAM Attention map $G$ (orange shades). **f** Same example sample as in E, showing the defected regions (nonhexagonal cells, $D = 1$, colored in orange) and the nondefected ones (hexagonal cells, $D = 0$, colored in gray). **g** Boxplot of correlation between Grad-CAM attention $G$ and cell defects $D$ in the variable-disorder dataset, for low ($s^2 \in [0.25, 0.5)$), medium ($s^2 \in [0.5, 0.75)$) and high ($s^2 \in [0.75, 1]$) disorder silica samples. The panel shows that, particularly for low-disorder samples, the ResNet model is effectively focusing on the disordered regions in order to predict the disorder level of the samples, as expected. The scalebar of the images of silica is 10 Å.

image that are most relevant to differentiate one class from the other.

The example shown in Fig. 3e highlights that the ResNet is focusing on the structural defects in the configuration (i.e. cells that are not hexagonal), which we highlight in Fig. 3f where we color the non-hexagonal cells through a modified Voronoi construction as described in the Methods. A cross-correlation between the Grad-CAM attention maps and images with colored defects indicates a high Pearson correlation coefficient that decreases for large disorder (see Fig. 3g)). As illustrated in Fig. 1, Grad-CAM maps can be produced for different layers of the ResNet, corresponding to different resolutions $r$ for the images. The best correlation with structural defects is found at higher resolutions ($r = 32$, Supplementary Fig. 4).

**Residual neural networks predict rupture strain.** Next, we train the ResNet to predict the rupture strain given the image of the undeformed configuration. The results indicate again a high prediction accuracy for weakly disordered samples (Fig. 4a) which

then slightly decreases as the disorder level increases (Fig. 4b). We have also investigated how the quality of the prediction degrades as a function of the quality of the training images (see Supplementary Fig. 5). We then exploit again Grad-CAM to show that for the low disorder, accurate rupture strain predictions are obtained by the ResNet by focusing its attention on the region where rupture will actually initiate (see Fig. 4c). This is remarkable since the ResNet was not provided with any information regarding failure location and yet it can predict it after training with rupture strains only. We notice, however, that when attention maps become less localized (Fig. 4d), the prediction accuracy decreases.

In general, we observe that a high degree of localization of the Grad-CAM map is typical of samples with low rupture strain, while delocalized maps are found for samples with high rupture strain (Fig. 4e). If we compare the peak of the attention map with the true rupture location, we find a small location error for small rupture strains which increases with the rupture strain (Fig. 4f). The maps in Fig. 4c and d are obtained at a coarse resolution ($r = 4$). We also inspect Grad-CAM maps at a fine-grained

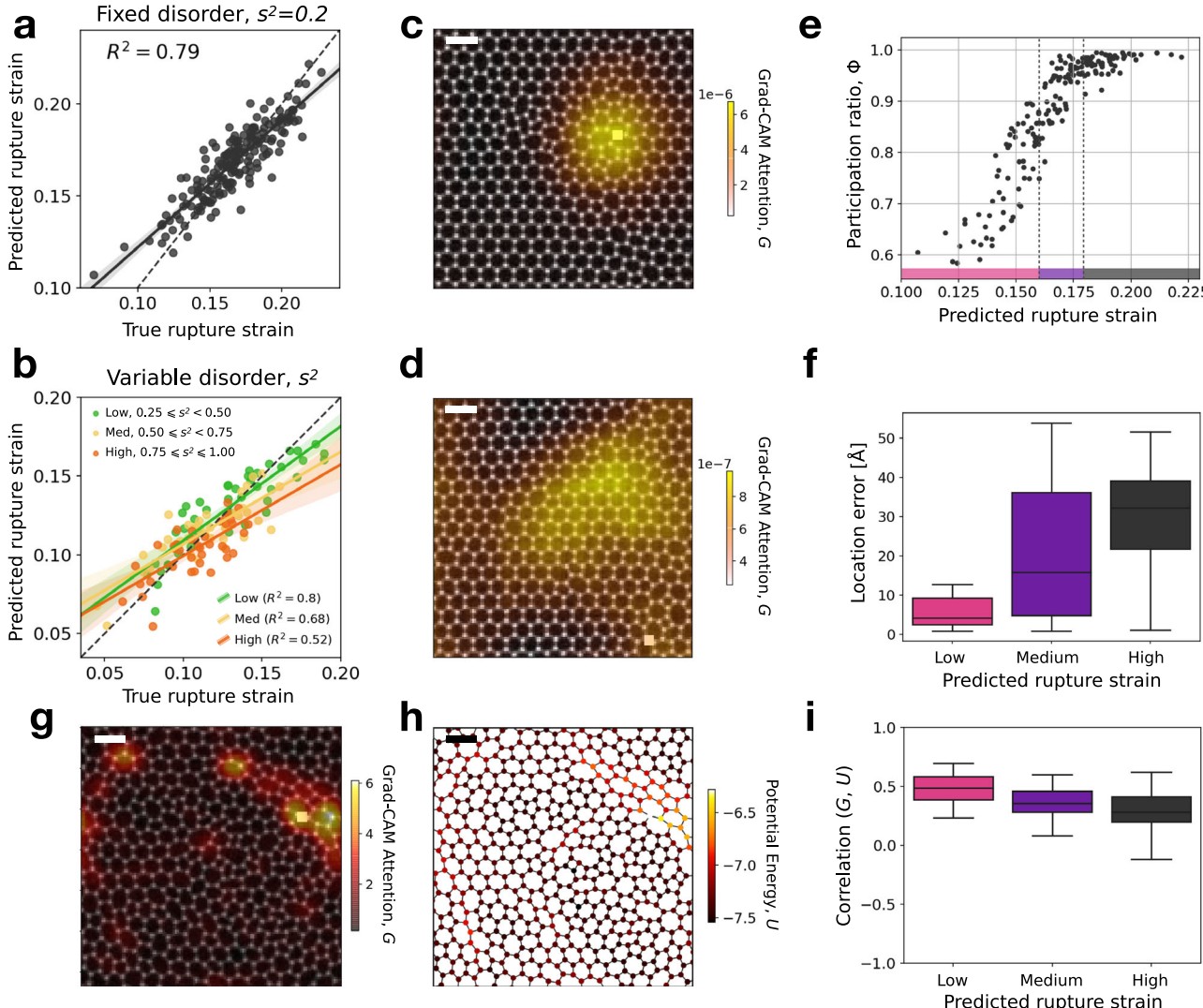

**Fig. 4 Machine learning predicts rupture strain. a** Scatter plot of the true rupture strain (defined as the first bond-breaking event) versus the prediction of the ResNet model. Black dots correspond to silica samples generated at a fixed disorder level of $s^2 = 0.2$. A linear fit yields $R^2 = 0.79$. **b** Same, for silica samples generated at variable disorder levels $s^2 \in [0.25, 1]$, grouped into three classes for simplicity. **c** Example of a low rupture strain silica sample (black-and-white image) superimposed with the Grad-CAM attention heatmap $G$ (transparent-to-yellow shading) derived from the strain-trained ResNet model. The white square indicates the true rupture location. **d** Same, for a silica sample with high rupture strain. In this case, high Grad-CAM attention values $G$ (bright yellow regions) do not correspond to the true rupture location (white square). **e** Scatter plot of participation ratio $\phi$ versus predicted rupture strain showing that samples that break at low strain have a localized Grad-CAM Attention heatmap (low $\phi$) while those that break at high strain show a more globalized heatmap (high $\phi$). The pink, purple and gray lower bands indicate the regions of low, medium and high predicted rupture strain used in panels **f**, **i**. **f** Boxplot of location errors, defined as the distance between the highest intensity point of Grad-CAM heatmap and the true rupture location; for different levels of the predicted rupture strain following the coloring of panel **e**. **g** Example of silica sample (black-and-white image) super-imposed with Grad-CAM attention $G$ (black-red-yellow colorscale). **h** Same sample, with atoms colored according to their potential energy $U$. **i** Correlation between Grad-CAM attention $G$ and potential energy $U$, over the entire dataset, for different levels of predicted rupture strain. The scalebar of the images of silica is 10 Å.

resolution ($r = 32$) (Fig. 4g) and find they correlate with the local potential energy $U$ of the configurations (see Fig. 4h), particularly in samples with low rupture strain (Fig. 4i).

**Residual neural networks predict location of rupture and crack path**. We then train our ResNet on the dataset at fixed disorder to predict the spatial location of rupture (first bond breaking). To this end, we train the ResNet separately using the $x$ and $y$ coordinates, which produces accurate predictions (Fig. 5a). In order to understand which features of the input images are most relevant for predicting the rupture location, we combine the information of the two Grad-CAM maps $G_x$ and $G_y$ to obtain a vector field

$(G_x, G_y)$ as shown in 5b). We find that the vector points towards the rupture location and that the divergence field produces band regions parallel to the crack direction. We calculate the error for each location prediction $e_r$ and plot its cumulative distribution for three different levels of rupture strain. We note that again the prediction is more accurate for samples that break at low rather than at medium and high rupture strains (see Fig. 5b and Supplementary Fig. 6).

As a further target for the ResNet, we consider the complete crack path which we represent as an image (see Methods). There is a strong correlation between the location of the first bond breaking and the average position of the final path (see

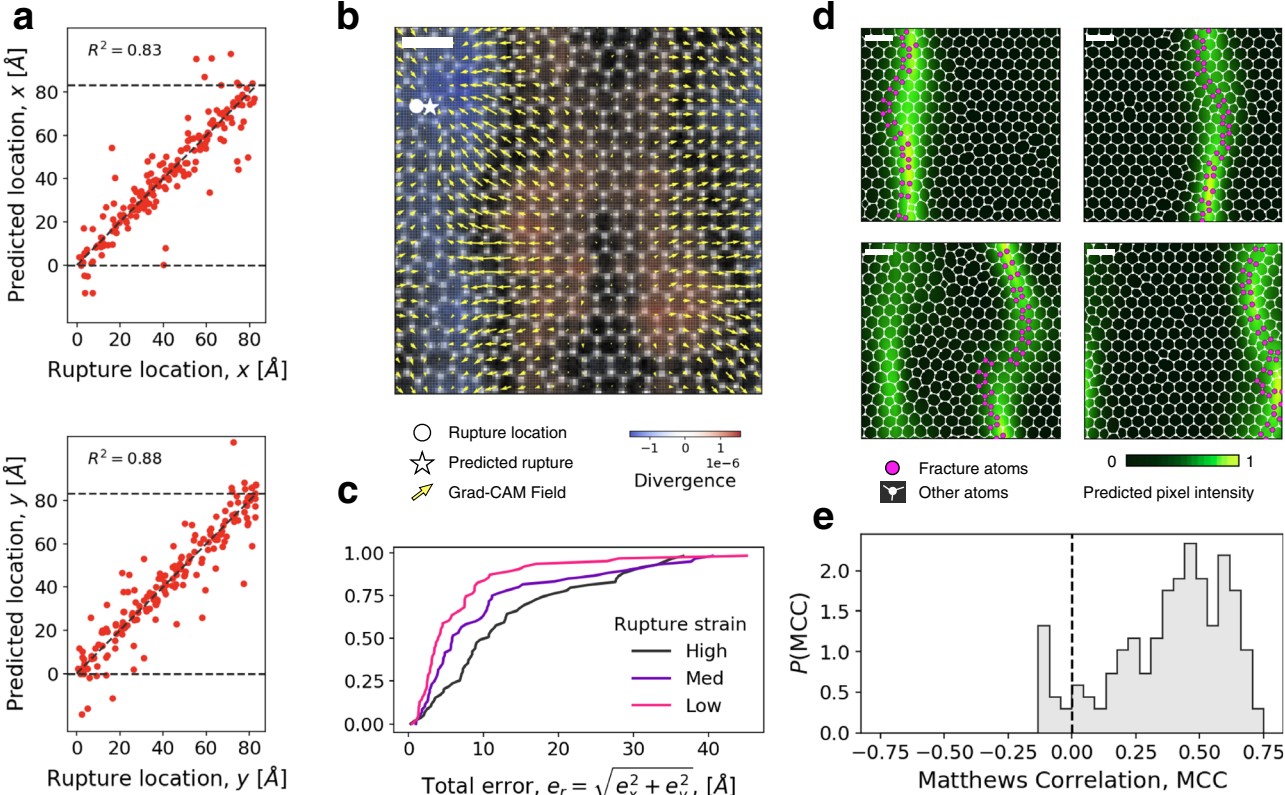

**Fig. 5 Machine learning predicts rupture locations. a** ResNet prediction matches the true rupture location on $x$ and $y$ coordinate (top and bottom panels respectively) in stretched silica glass samples. Rupture location is defined as the location of the first breaking bond. **b** Grad-CAM attention field $G_x, G_y$ obtained from the two models shown in panel A: the first component of the vectors $G_x$ represents the Grad-CAM attention value for the $x$ prediction, whereas the second is the value $G_y$ corresponds to the $y$ prediction. The overlapped colormap reports the divergence of the field, from negative (blue) to positive (red) values. The circle indicates the true rupture location of the sample, while the star marks the predicted one. **c** Cumulative distribution of the total location error $e_r$ for the three different rupture strain levels defined in Fig. 4e. **d** Examples of full crack path prediction. The predicted pixel intensity is shown as a black to green shaded background. The unstretched silica is overlayed in white and the atoms involved in the actual crack path are colored in bright magenta. **e** Distribution of the Matthews correlation coefficient (MCC) across the entire test set. The scalebar of the images of silica is 10 Å.

Supplementary Fig. 7), but the crack path is not always straight. We use an image-to-image algorithm inspired by colorization models[37] so that the ResNet learns the correspondence between an image of the undeformed configuration and an image of the crack obtained after stretching (see Supplementary Fig. 8 for a schematic of the ResNet used). Figure 5d shows four examples of model predictions (green shaded areas) overlaid with images of the unstretched configuration where the atoms involved in the crack are highlighted in magenta (see Supplementary 9 for more examples). In all cases, the prediction matches very well the actual crack path. To quantify the accuracy of the predictions, we compute the Matthews correlation coefficient (MCC) as explained in the Methods section (Fig. 5e).

**Transfer learning.** Having demonstrated the capabilities of our ResNet in predicting a series of characteristic features of the failure process, we show how these results can be generalized. Ideally we would like to train the ResNet with a data set and make predictions on slightly different types of data. Consider for instance the issue of predicting the fracture strain and the rupture location for a sample that is considerably larger than the samples used for training. This is important since artificial neural networks have memory limitations associated with the size of the images used for training. To illustrate this point, we make predictions on samples four time larger than those used for testing (see Fig. 6a) We then slide a square window over the entire

sample and predict the rupture strain for each position of the sliding window, defined as the center of the window. The predicted strain is not uniform along the sample and we conjecture that regions of low predicted rupture strain are more likely to contain the actual rupture location. The correctness of this assumption can be clearly seen in the examples shown in Supplementary Fig. 9. We then select the fraction of the sample with lowest predicted strain (green shade) and compute the probability that the actual fracture location is included within the selected area. Selecting only 25% of the area, the probability of a correct prediction is close to 75% (see Fig. 6b).

We can employ a similar *transfer learning* strategy to data extracted from experimental TEM images of 2D silica glasses [23] available from the literature (see Fig. 6c–e). Since fracture tests were not performed experimentally, we use the ResNet model that was trained to predict disorder and apply it to experimental data. Figure 6c–e displays the associated Grad-CAM attention maps that highlight again the topological defects. The predicted values of the disorder are also in good agreement with direct measurements of the ring size distributions. We also use the experimental data to perform SVM classification and display in Fig. 6c–e the atom that are likely to deform irreversibly upon strain in the horizontal direction. Notice that the atoms predicted to be prone to failure according to the SVM are clustered in area of high attention according to the Grad-CAM. Similar results are obtained considering strain in the vertical direction.

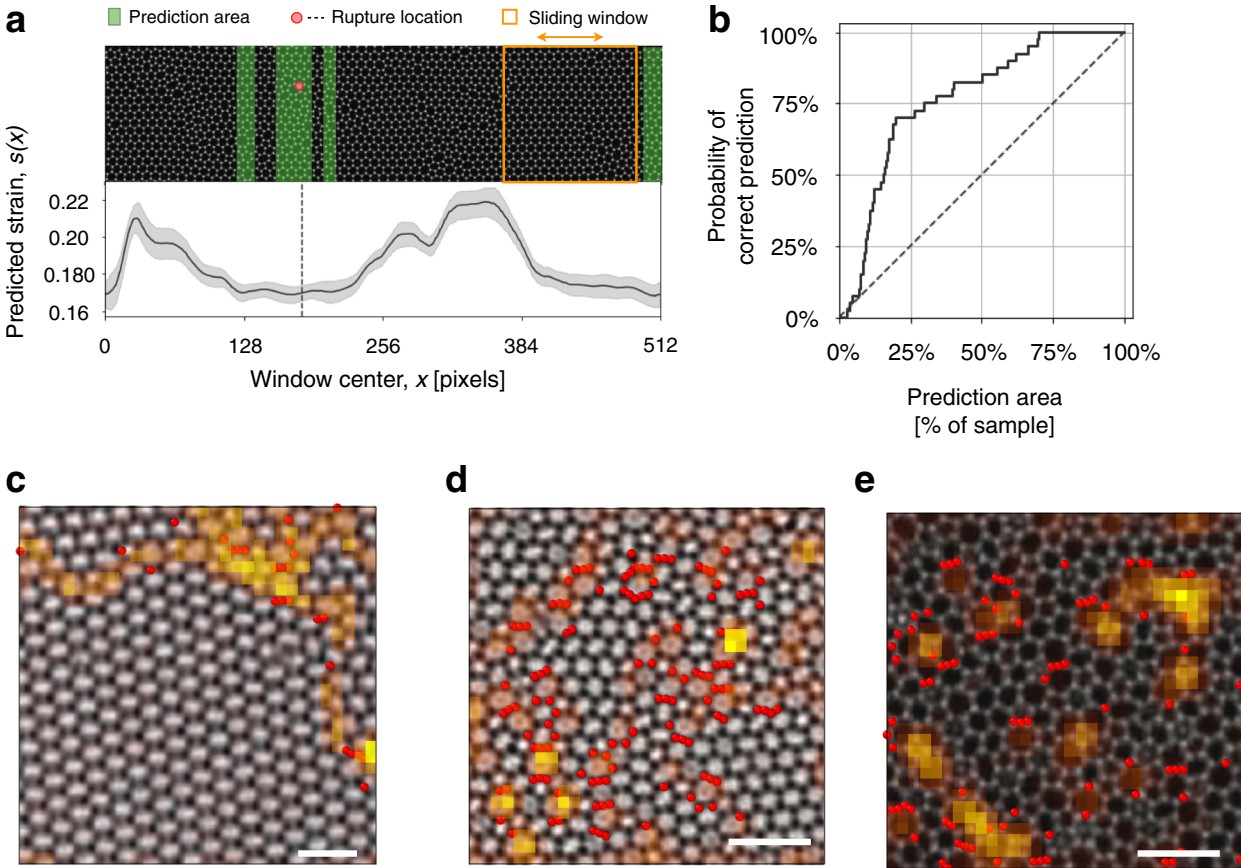

**Fig. 6 Transfer learning to tackle larger samples and real experiments. a** Transfer learning on sample of larger size. The ResNet model, originally trained on 128x128 pixels samples to predict rupture strain, is used to predict the rupture strain of different regions of the shown larger silica sample, of size 512x128 pixels. When this sample is stretched by MD it produces the rupture event (bond breaking) indicated by the red dot and vertical dotted line. In this example, the rupture location lies inside the green prediction area, a region of lowest predicted strain. The predicted strain is computed by sliding the orange window over the whole sample. The shaded grey band corresponds to the standard deviation of the rupture strain computed over all the possible images obtained from the same configuration by data augmentation. **b** Probability of correct prediction (finding the rupture location inside the prediction area) as a function of the size of the prediction area. The diagonal dashed line corresponds to a random guess. The model is able to identify the rupture location with accuracy well beyond the random guess baseline. **c**–**e** Experimental TEM images of two-dimensional silica glass samples from[23] are analyzed by the ResNet model trained on simulated data to predict disorder and by SVM classification. Grad-CAM attention maps $G$ highlight the topological defects. Atoms predicted by SVM to be deforming irreversibly after straining in the horizontal direction are reported in red. Scalebars are 20 Å. Reproduced with permission (**c**) American Chemical Society.

## Discussion

In conclusion, we have exploited different machine learning strategies to predict the failure behavior of silica glasses. Our results highlight the limitations of commonly used atom-level local models that in the case of fracture unavoidably give rise to a large number of false-positive predictions, but have clear advantages in terms of algorithmic scalability. Global image-based models provide instead reliable predictions and thanks to Grad-CAM maps allow us to visually inspect the structural determinants of failure and to correlate them with physical and topological signatures of the atomic microstructure. Our results illustrate that it is possible to link the failure properties of glasses to their structure by using machine learning models trained on large-scale atomistic simulations. In particular, it is possible to reliably identify 'zones of interest' where cracks are likely to nucleate and propagate. This may pave the way for future hybrid multi-scale simulation schemes which combine numerical efficiency with high accuracy: The capability to a priori identify those regions of a sample (here rupture site and fracture path) which require a physically accurate description allows to use hybrid simulation schemes which combine such accurate descriptions

with coarser ones of the rest of the sample, such as to significantly reduce numerical cost without compromising numerical accuracy.

## Methods

**Generation of glassy configurations**. Glassy configurations are generated in two steps: First, a two-dimensional network of Si-only atoms with pre-fixed target ring-size distribution is created using spring-like potentials in dual space. Then, the full silica bilayer is formed and relaxed using the Watanabe potential[31]. The disorder of the silica sample is therefore controlled by a single parameter $s^2$, the variance of the ring-size distribution, which we call simply "disorder level" throughout the manuscript. We create three datasets of 1000 configurations each with three different levels of disorder: two at a fixed disorder levels $s^2 = 0.2$ and $s^2 = 0.3$, and one with varying levels of disorder, with $s^2$ uniformly distributed between 0.25 and 1. Below we include details of the two-step procedure. The variable disorder dataset has been divided into three groups of increasing disorder level, labeled as Low ($s^2 \in [0.25, 0.5)$), Medium ($s^2 \in [0.5, 0.75)$) and High ($s^2 \in [0.75, 1]$) along the manuscript. The fixed disorder dataset, instead, has been divided according to the fracture strain into three groups of the same size. Since in this case, the distribution of fracture strain is not uniform, the obtained intervals are of unequal bin size, and have been labelled as Low ($\epsilon < 0.16$), Medium ($0.16 \leq \epsilon \leq 0.18$), and High ($\epsilon > 0.18$).

We use a Monte Carlo dual-switch procedure[27,30] to generate a two-dimensional network representing Si atoms with a preset ring-size distribution and nearest-neighbour ring-size correlations (known as Aboav-Weaire law, the

experimetally observed tendency of large rings to be close to small rings). Following[27,30], the position of the Si atoms is adjusted by minimizing a spring-like potential in the dual space of ring-to-ring interactions after each Monte Carlo step. We use a Monte Carlo temperature of $10^{-4}$ and $N = 10^4$ Monte-Carlo steps and a value of $\alpha = 1/3$ for the Aboav-Weaire law, which corresponds to experimental observations[27].

After the above preparation, oxygen atoms are added midway each Si-Si bond, and the resulting structure is then isotropically rescaled so that the average Si-O bond-length becomes close to the rest length of $\sim 1.65$ Å. The 2D silica is finally formed by duplicating the so-obtained layer and by connecting the Si atoms in the two layers with oxygen atoms, see Fig. 1a. Each structure typically consists of 3456 atoms arranged in a box of side $L \sim 85$ Å, slightly variable with the amount of disorder. Periodic boundaries were applied along the $x, y$ directions, corresponding to the silica sheet plane. Finally, the whole configuration is relaxed using the Watanabe potential[31], see the details in the following section. All the simulations were performed using LAMMPS[38]. We then calculate the coordination number for each atom using a cutoff of 2.2 Å, and verify that it is equal to 4 for Si atom and 2 to for O atom. We discard from the analysis the few samples where some atoms are incorrectly coordinated. This ensures that all the bonds are of the Si-O-Si type.

**Interatomic potential**. The Watanabe potential for the silica class has the advantage of implicitly replacing the usual Coulomb interaction term with a coordination-based bond softening function for Si-O atoms that accounts for environmental dependence, which is of particular importance for surface effects, which are prominent in quasi-2D systems.

The general form of the potential consists of two terms: a two-body interaction that depends on distance and a Stillinger-Webber-like three-body interaction. Specifically, the total potential energy $\Phi$ is written as:

$$\Phi = \sum_i \sum_{j>i} \epsilon f_2 + \sum_i \sum_{j\neq i} \sum_{k>j, k\neq i} \epsilon f_3 \tag{1}$$

The two-body interaction term between the $i$-th and $j$-th atom is given by:

$$f_2(r_{ij}) = g_{ij} A_{ij} \left( \frac{B_{ij}}{r_{ij}^{p_{ij}}} - \frac{1}{r_{ij}^{q_{ij}}} \right) \exp\left( \frac{1}{r_{ij} - a_{ij}} \right) \tag{2}$$

where $r_{ij}$ is the distance between the $i$-$j$ pairs. The detailed parameter values are reported in ref. [39]. $g_{ij}$ is the softening function depending on the coordination numbers of the $i$ and $j$ atoms:

$$g_{ij} = \begin{cases} g_{Si}(z_i) g_O(z_j) & i = Si, \ j = O \\ g_{Si}(z_j) g_O(z_i) & i = O, \ j = Si \\ 1 & \text{otherwise} \end{cases} \tag{3}$$

and

$$g_{Si}(z) = \begin{cases} (p_1 \sqrt{z + p_2} - p_4) \exp[p_3/(z-4)] + p_4 & z < 4 \\ p4 & z \geq 4 \end{cases} \tag{4}$$

$$g_O(z) = \frac{p_5}{\exp[(p_6 - z)/p_7] + 1} \exp[p_8(z - p_9)^2] \tag{5}$$

where $z_i$ and $z_j$ are the coordination numbers of atoms $i$ and $j$. The coordination number $z_i$ is defined by a cutoff function $f_c$:

$$z_i = \sum_{j \text{ s.t. } i \neq j} f_c(r_{ij}) \tag{6}$$

with

$$f_c(r) = \begin{cases} 1 & \text{if } r < R - D \\ 1 - \frac{r-R+D}{2D} + \frac{\sin[\pi(r-R+D)/D]}{2\pi} & \text{if } R - D \leq r < R + D \\ 0 & \text{if } r \geq R + D \end{cases} \tag{7}$$

with $R$ and $D$ parameters.

The three-body interaction term depending on the positions of $i$-th, $j$-th, and $k$-th atoms, has the following form:

$$f_3(r_{ij}, r_{ik}, \theta_{jik}) = \Lambda_1(r_{ij}, r_{ik})\Theta_1(\theta_{jik}) + \Lambda_2(r_{ij}, r_{ik})\Theta_2(\theta_{jik}) \tag{8}$$

where $r_{ik}$ is the distance between $i$-$k$ pairs. $\theta_{jik}$ is the bond angle between $r_{ij}$ and $r_{ik}$. For $n = 1, 2$ we can write

$$\begin{aligned} \Lambda_n &= \lambda_{n,jik}(z_i) \exp\left[ \frac{\gamma_{n,jik}^{ij}}{r_{ij} - a_{n,jik}^{ij}} + \frac{\gamma_{n,jik}^{ij}}{r_{ik} - a_{n,jik}^{ik}} \right] \\ \Theta_n &= (\cos\theta_{jik} - \cos\theta_{n,jik}^0)^2 + \alpha_{n,jik}(\cos\theta_{jik} - \cos\theta_{n,jik}^0)^3 \\ \lambda_{n,jik}(z) &= \mu_{n,jik}(1 + \nu_{n,jik} \exp[-\xi_{n,jik}(z - z_{n,jik}^0)^2]) \end{aligned} \tag{9}$$

**Straining of samples**. After an initial atomic relaxation of the silica structures (see Fig. 1b left panel), in which the cell vectors length was allowed to vary in order to minimize pressure, fracture tests were simulated by performing iterative expansion and relaxation, via the athermal quasistatic (AQS) protocol as following. The silica

structure was expanded in the $x$ direction by $\Delta x = 5 \cdot 10^{-5} L_x^0$, with $L_x^0$ the initial size of the box along $x$. Subsequently, a damped dynamics with viscous rate of $1$ ps$^{-1}$ was performed until the maximum force was below the threshold of $10^{-8}$ eV/Å. Such procedure was repeated while monitoring the potential energy of each atom in order to detect any drop $> 0.1$ eV, corresponding to a bond breaking. The potential energy per atom is computed considering its interaction with all other atoms in the simulation. When the energy contribution is produced by a set of atoms (e.g., 3 atoms for the 3-body interaction term), that energy is divided in equal portions among each atom in the set. For a covalent system subject to strain, the potential energy per atom must always grow with the strain unless some bond breaking occurs. We also verify that for the atom associated to the first bond breaking, in the terms discussed above, the distance from its nearest neighbors suddenly changes after the energy drop. In particular, we check that the O atom remains bound to a single Si atom, with a bond length that we verify to be $< 1.7$ Å, corresponding to the rest length, while the other Si is located at much larger distance, $> 2.2$ Å. The fact that an O atom, initially equidistant between two Si atoms suddenly moves towards one of the two is an unequivocal sign of bond breaking. We refer to that situation whenever the 'bond breaking' concept is recalled in the text.

Configurations were saved in the correspondence of the first bond breaking (see Fig. 1b central panel) and in the correspondence of the full fracture formation (see Fig. 1b right panel).

**Generation of images of silica configurations**. To generate images from the silica configurations for the machine learning tasks, we add Gaussian noise at the Si atom positions and a uniform background noise over all the sample, to then create a two-dimensional heatmap of the required pixel dimensions, in our case 128 pixels per side. In this images, oxygen atoms are barely visible, but this should not be relevant since our lattices are constructed in a way so that all the Si atoms were 4-coordinated, and therefore adding the O atoms in-between the Si-Si bonds would not add any new structural information.

**Data augmentation**. We use standard data augmentation techniques on our generated images to increase the sample size of our datasets. Beyond the standard horizontal and vertical flips, we leverage the periodic boundary conditions (PBC) of the system under study, which allow us to use translations over PBC as well. To be precise, we apply 64 random translations over PBC to each original image. Of these, 32 are flipped horizontally and 32 are flipped vertically (so that 16 are flipped both vertically and horizontally). The data augmentation has two advantages: first it increases the number of images we can feed to the machine learning algorithms, which otherwise would be a limitation since AQS simulations are computationally expensive; and second, it allows to average predictions over the 64 copies of each image, taking care of the inverse transformations if the prediction is a position. We denote averages of predictions over data augmentation as $\langle \cdot \rangle_{DA}$. Taking the average of predictions over data augmentation can be seen as a form of noise cancelling trick that leads to more robust predictions.

**Reconstruction of silica configurations from experimental TEM images**. We reconstruct a silica glass configuration from the experimental TEM image shown in Fig. 6c–e, taken from ref. [23], as follows:

1. We manually select a region of interest where the Si atoms can be clearly seen.
2. We use the trackpy python package to detect the position of Si atoms, since they are brighter than O atoms. In particular, we use the function trackpy.locate with a diameter parameter of 11.
3. We manually add (remove) missing (spurious) Si atoms.
4. We infer the bonds using a modified version of the Voronoi diagram that takes into account the coordination number of Si atoms.

In this way, from an experimental TEM image we obtain a collection of Si atom positions and pairs of atoms that form bonds, which can be used in molecular dynamics simulations as well as for our Grad-CAM analysis, as shown in Fig. 6c–e.

**Voronoi-like method**. In order to quantify the spatial distribution of defected cells in the silica lattice, we propose an algorithm that is able to automatically visualize those particular cells starting only from the collection of the coordinates of the Si atoms. In particular, this algorithm exploits the Voronoi diagram and the Delaunay triangulation. At first we build the Voronoi diagram for the set of the Si atoms coordinates. Taking into account the fact that the coordination number for Si is three in a silica monolayer, we notice that a Si atom shares the three widest sides of its Voronoi cell with the three Si atoms which are physically bonded to it. This allows to reconstruct all the physical Si-Si bonds in the silica configuration. Then we build the Delaunay triangulation for the set of the Si atoms coordinates. Some of the sides of those triangles are the physical Si-Si bonds. We notice that an ensemble of triangles, which is enclosed only by physical Si-Si bonds and which does not contain any of the physical Si-Si bonds inside it, is a silica lattice cell. So, merging those triangles with an iterative process, we find the list of Si atoms, which realize each real lattice cell. This knowledge allows us to recognize which cells are defected or not.

**Machine learning models**. We use custom architectures based on residual neural networks (ResNet)[40] and image-to-image algorithms inspired by colorization models[37] to tackle four different learning tasks: disorder learning, rupture strain learning, rupture location learning and full crack path learning. In what follows, we list technical details of each architecture and associated parameters, as well as details on the training procedure.

For the disorder, rupture strain and location learning tasks we use a ResNet50 model (Supplementary Fig. 3) as implemented in the keras python library modified to perform regression instead of classification, as explained in[41]. The modification, in short, consists in substituting the last standard "softmax" layer typical of classification tasks by a fully dense layer, allowing the model to be trained on regression tasks. ResNet50 is a very complex deep convolutional neural network architecture, comprising many kinds of layers and skip connections. However, the size of convolutional filters remains constant through a set of layers, named 'convolutional blocks', the name assigned by the keras library to each layer starts with the relative convolutional block. In the present case of study, the input image has a size of 128x128 pixels. Notice that ResNet50 can handle any linear input size bigger than 32 pixel. First, the image is elaborated and transformed in filters of size 64x64. Then those filters are processed through the convolutional blocks and are transformed in filters of 32x32, 16x16, 8x8 and 4x4 dimensions. Data is randomly split into training set (72% of data), validation set (8% of data) and test set (20% of data). Data augmentation is performed after data splitting, which leads to a total of 41,792 images for the train test, 4,608 images for the validation set and 11,584 images for the test set for the fixed disorder $s^2 = 0.2$ dataset. For the case of variable disorder $s^2 \in [0.25, 1]$, we work with a total of 27,968 images for the train set, 3,136 images for the validation set and 7 808 images for the test set. In all three tasks we train the model for 100 epochs using the ADAM optimizer, saving the validation loss at each step, and keep the model weights of the epoch with lowest validation loss. We do not tune any additional parameter. All figures shown in the manuscript correspond to the test set, unused during training. The loss function is a simple mean squared error for the disorder and strain test. The location prediction, however, requires a custom loss function $\mathcal{L}$ that has an additional bias term and that takes into account periodic boundary conditions when computing distances. In particular, we use a two-term loss function $\mathcal{L} = \mathcal{L}_1 + \mathcal{L}_2$, where the first term

$$\mathcal{L}_1 = \left\langle \|y - \hat{y}\|^2_{\text{PBC}} \right\rangle \tag{10}$$

is the mean squared error between the targets $y$ and the predictions $\hat{y}$, and $\| \cdot \|_{\text{PBC}}$ is an Euclidean norm taking into account the periodic boundary conditions of the system, that is, along both the $x$ and $y$ coordinates. The second term

$$\mathcal{L}_2 = \left( \langle y \rangle - \langle \hat{y} \rangle \right)^2 \tag{11}$$

is an overall bias term, the (squared) difference between the average target and the average position. In practice, we have observed faster model convergence when adding this term to the loss function.

For the fracture path prediction task, instead, we use an image-to-image model inspired by image colorization algorithms[37]. To be precise, we couple the ResNet50 model with upsampling and convolutional layers, see Supplementary Fig. 8 and 9 for details. In summary, the image-to-image model starts with a 128x128 pixels image, has a central core of 4x4 pixels with 2048 filters, and then grows again to build the target 128x128 pixels image. The target images are a noisy version of the input silica image where only atoms belonging to the crack path are shown. The noisy modification consists in applying a Gaussian filter of standard deviation $\sigma = 4$ in the $x$ and $y$ directions, and then adding random uniform noise in the range (0, 0.25). Supplementary Fig. 9 shows some examples of noisy targets and associated predictions. The use of noisy targets avoids that the image-to-image model to concentrates on the uniform background (most of the image) instead of the fracture atoms, as would happen otherwise.

Additional to the ResNet approach, we use support vector machines (SVM) to predict the atoms involved in the first bond break. The SVM gets fed a vector of symmetry functions encoding the neighborhood of an atom to decide wether it is involved in the first bond break. The rational behind this approach is to see how good one can perform in this setting with a simple model and straightforward physics-guided local descriptors. The descriptors are radial symmetry functions calculated from the undeformed initial configuration as well as the affine transformed initial configuration in order to incorporate the underlying physical symmetries. The affine transformation is applied by scaling the atoms with the cell in order to mimic the loading. Without this transformation the symmetry functions are entirely rotation invariant which is wrong in the context of mechanics, as the local atomic neighborhood is highly anisotropic with regards to different load directions. As the SVM makes prediction for a single atom and receives information about the local neighborhood of a single atom, it may end up suggesting more than two atoms within the same simulation box to undergo bond breaking. In plain words, this estimator has no concept of an atomic system, just single atoms.

**Grad-CAM attention**. Grad-CAM was first introduced in[29] in order to understand the decisions made in Convolutional Neural Networks (ResNet) with visual explanations. The basic idea of Grad-CAM is to highlight which parts of an image are of importance to obtain the prediction. Let us summarise here how Grad-CAM

works. Denoting the input image as $\vec{\gamma}$, the target as $t$, and the whole ResNet simply as $F(\cdot)$, we can write:

$$t = F(\vec{\gamma}) \tag{12}$$

The target $t$ can represent both a qualitative or a quantitative variable, depending on the problem at hand. While Grad-CAM was first introduced and applied to classification problems (where $t$ would be a qualitative variable), in the present work we modify the standard Grad-CAM algorithm to deal with regression problems. Therefore, in what follows $t$ represents a quantitative variable and is a scalar quantity. Let $f^k_{ij}(\vec{\gamma})$ the $i, j$ pixel of the filter $k$ of a certain convolutional layer in the ResNet. Usually one takes $f$ as the last convolutional block, since it tends to collect the most important features, but the following treatment can be extended to any convolutional block of the ResNet. We can define the global importance $I^k$ of the filter $f^k$ for the prediction $y$ as:

$$I^k = \frac{\sum_{ij} \frac{\partial t}{\partial f^k_{ij}}}{Z} \tag{13}$$

where the gradient is obtained with back-propagation and $Z$ is a normalization constant. Summing over all filters and averaging over data augmentation, we obtain the Grad-CAM Attention for a given pixel $(i, j)$:

$$G_{ij} = \left\langle \left| \sum_k I^k f^k_{ij} \right| \right\rangle_{\text{DA}} \tag{14}$$

Throughout the manuscript we also use $G$ to denote Grad-CAM Attention of an unspecified pixel, omitting the $(i, j)$ sub-index for simplicity.

*Resolution of Grad-CAM attention heatmaps*. A Grad-CAM Attention heatmap $G$ has an associated resolution value $r$, which depends on the convolutional block being used: The Grad-CAM heatmap from the first convolutional block is an heatmap of size 32x32, the Grad-CAM heatmap from the second convolutional block is an heatmap of size 16x16 and so on. In order to highlight the important part of the input image (the configuration image of size 128x128), we have resized the Grad-CAM heatmaps to a size of 128x128 pixels and superimposed them on the input image. For instance, when using the last convolutional block to compute $G$ we obtain a heatmap $G_{ij}$ of $4 \times 4$ possibly different values, so the Grad-CAM resolution in that case is $r = 4$ (the resolution of the Grad-CAM heatmap coincides with the dimensions of the output of the relative convolutional block, see Supplementary Fig. 3 for details). Figure 1d shows the resolution of some example Grad-CAM heatmaps, from $r = 32$ to $r = 8$. Supplementary Fig. 4 shows correlation coefficients between cell defects $D$ and Grad-CAM attention values $G$ computed at different resolution levels $r$, from $r = 32$ (first convolutional block) to $r = 4$ (last convolutional block).

*Grad-CAM attention fields*. The Grad-CAM attention *field* shown in Fig. 5b, $(G_x, G_y)$, is a two-dimensional vector field build from the Grad-CAM attention values $G_x, G_y$ of two independent ResNet models: one was trained on the $x$ component of the rupture location and the other trained on the $y$ component.

**Definition of participation ratio**. We define the participation ratio $\phi$[42] of a Grad-CAM map $G_{ij}$ as

$$\phi = \frac{\left[ \sum_{i,j=1}^N G^2_{ij} \right]^2}{N^2 \sum_{i,j=1}^N G^4_{ij}} \tag{15}$$

where $G_{ij}$ is the Grad-CAM attention value of pixel $(i, j)$ and $N^2$ is the number of pixels of the image, (in our case $N = 128$). The participation ratio, in this context, can be understood as a measure of "globalization" of the attention map, so that low $\phi$ values correspond to very localized attention maps $G_{ij}$, that is, to cases where the Grad-CAM model focuses on a particular region of the image to make its predictions; and conversely high values of $\phi$ correspond to cases of high globalization, where the model needs to make use of the entire image to reach a prediction.

**Correlation coefficients**. The correlations between Grad-CAM attention $G$ and cell defects $D$ shown in Fig. 3g and Supplementary 4 are computed as the Pearson product-moment correlation coefficient across pixels. That is, for each pixel $(i, j)$ we associate a Grad-CAM attention value $G_{ij}$ and a cell defect value $D_{ij}$, which is 1 if the pixel is part of a defected cell, and 0 otherwise. The correlation between Grad-CAM attention $G$ and potential energy $U$ shown in Fig. 4i, instead, is computed at the atom level, where for each atom we associate a potential energy $U$ and a Grad-CAM value $G$ at the corresponding location.

To quantify the agreement between the prediction of the colorization model and the fracture atoms in the crack path prediction task, we first need to threshold the image prediction to obtain a per-atom binary prediction. That is, each atom is either a fracture atom or not (ground truth), and is either predicted as fracture or not. The threshold is chosen so that atoms that lie on the top 10% prediction intensity (brightest green area) are classified as fracture atoms. Then, the Matthews correlation coefficient is computed as implemented in the scikit-learn python package.

**Symmetry functions and support vector machines**. We compute the radial symmetry functions

$$\psi_i(\mu) = \sum_j^N e^{(r_{ij}-\mu)^2/\delta^2} \qquad (16)$$

for particles of the initial undeformed configuration and for particles of an affine transformed configuration according to the deformation gradient. The off-diagonal elements of the deformation gradient are zero while the diagonals are $(1+\epsilon, 1, 1)$ since we apply uniaxial strain to our samples. The affine deformation causes this approach to be sensitive to the orientation of the atomic neighborhood with respect to the external loading while remaining independent to translation and rotation. We calculate the symmetry functions separately for each particle type relation (Si-Si, Si-O, O-O) as is common in this approach[16]. To perform predictions with these newly created features, support vector machines are used.

## Data availability

The data generated in this study have been deposited in the zenodo database under accession code [https://zenodo.org/deposit/6335037].

## Code availability

The codes used in this paper are available at https://github.com/ComplexityBiosystems/2D-silica-ML.

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

## Acknowledgements

We acknowledge support from the Deutsche Forschungsgemeinschaft (DFG, German Research Foundation) Grant no. 1 ZA 171/14-1 (S.H., M.Zai and S.Z.). We acknowledge participation in the training programme of the FRASCAL graduate school (DFG, German Research Foundation) - 377472739/GRK 2423/1-2019 (S.H.).

## Author contributions

F.F.C., M.Zan, and S.H. analyzed data. S.B. and R.G. performed simulations. M.Zai and S.Z. designed and coordinated the study. All the authors wrote the paper.

## Competing interests

The authors declare no competing interests.
