## [Peer Review File · Nature Communications]

Title: Predicting the failure of two-dimensional silica glassesREVIEWER COMMENTS

Reviewer #1 (Remarks to the Author):

The authors have performed a machine learning study aiming at predicting the failure of two-dimensional silica glasses. Understanding the fracture resistance of disordered materials is an important challenge and the two-dimensional silica glass is a good model system for eventually understanding the behavior of more complex multicomponent glasses in three dimensions. Despite the complex structure of glasses at different length scales, the prospects of using machine learning, both in this and recent related studies, to predict the fracture response of glasses are exciting.

Indeed, the proposed machine learning algorithms in this study are promising, but there several questions and concerns that the authors should address, especially as it relates to the data acquired from molecular dynamics simulations. Therefore, I cannot recommend acceptance of the manuscript in Nature Communications in its current version.

1. I agree that the control of ring size distribution can impact the mechanical responses of silica glass, but it appears that all the data sets used for the machine learning calculations are on samples that exhibit brittle fracture behavior as shown in Figure 2. As reported in Ref. 27, ductile fracture behavior can also be observed in 2D silica glass. The authors therefore need to test if the method can also be used for a brittle-to-ductile transition. This would be an important test to include in a revised manuscript. In addition, some crystalline phases of 2D silica should also be included in the data sets to improve the generalization of the prediction.

2. The motivation for using the Grad-CAM method should be explained in more details. What are the pros and cons of this approach?

3. The authors should better explain the reason for using the Watanabe potential in simulating mechanical behavior of 2D silica. The mechanical properties of silica glasses seem to be largely overestimated as shown in Fig. 2 when compared to the results of the following references: 1) Gao, Z., Dong, X., Li, N. & Ren, J. Novel Two-Dimensional Silicon Dioxide with in-Plane Negative Poisson's Ratio. *Nano Lett.* 17, 772-777 (2017); 2) Zhang, J. Phase-dependent mechanical properties of two-dimensional silica films: A molecular dynamics study. *Computational Materials Science* 142, 7-13 (2018).

4. The relationship between the brittleness and disorder should be provided and discussed.

5. Analyses of rupture and fracture are based on the breakage of Si-O bonds, but oxygen atoms are invisible in the provided images. The authors should therefore discuss whether – and if so, to which extent – the oxygen atoms play a role in the machine learning fracture prediction.

6. It would be interesting if the author could address whether there is any relationship between the position of first rupture and the full fracture path. It may be helpful for illustrating whether these two

properties arise from the same structural origins.

7. As mentioned in the manuscript, both sides of the sample have been included in the data set. The authors should demonstrate whether the two sides of the sample can generate the same prediction.

8. For the transfer learning approach to be able to tackle large samples, I am wondering whether there is any dependence on the image scale? For instance, does the prediction still work well for an enlarged (zoomed-in) image?

Minor comments / typos:

9. In Figure 4, it is suggested to use the diagonal line to compute R^2 for accuracy comparison.

10. Page 15: "ensamble"

11. Page 16: the total of splitting of data is not equal to 100%.

Reviewer #2 (Remarks to the Author):

The manuscript "Predicting the failure of two-dimensional silica glasses" addresses a fundamental issue of the failure in disordered solids, attempting to connect peculiarities in 2D structure of silica glass with the observed specific failure strain, location of the first bond break event, and even crack path. Understanding such structure-properties relationships is an involved task in amorphous disordered structures partially due to the absence of classical defects. Because of that, machine learning techniques have become increasingly popular to predict the behavior of disordered solids. The authors address this in the introduction and apply both local (SVM) and global methods (ResNet + Grad-CAM) to study failure in 2D silica glass.

In my opinion, the most prominent new result that authors also highlight in the abstract is the employment of Grad-CAM to build attention maps associated with the failure location and crack path. Such Grad-CAM maps provide very good interpretability of the ML predictions and might be a promising tool for future studies in the related areas. Authors show a correlation between attention maps, disorder parameters, and failure location, although I would rather say these correlations are more of topological nature than physical. These particular results, in my opinion, provide enough novelty to consider publication in Nature Communications.

The manuscript relies on a well-established methodology to generate and strain atomic structures for further study, and I think that all necessary details are provided in the "Materials and Methods" part of the manuscript. The methods related to machine learning are also relatively standard (regarding

computer science rather than materials science). However, since these methods (SVM, CNNs, and Grad-CAM) are employed in a non-standard domain, they require much better explanation and visualization than provided in the current manuscript. I would like to emphasize that this is a major issue preventing the acceptance of the manuscript in its current form. I have an impression that authors prefer to demonstrate as many results as possible, somewhat sacrificing the clearness and completeness of the presentation.

To improve the manuscript, authors should address the following issues (in no particular order):

1. Please provide an exact architecture of ResNet50 used in this study. On page 19, the authors mention a 4x4 heatmap that corresponds to the last convolutional block and a 32x32 convolution block that corresponds to the first convolution block. It is unclear why these exact dimensions are used without seeing ResNet50 architecture in the manuscript. Moreover, even after analyzing classical ResNet50, I do not understand where these numbers come from. ResNet50 uses 224x224x3 images as input, while in the presented manuscript, the input dimensions are 128x128x1. Please, clarify.
2. As a continuation of issue #1, a more rigorous definition of the term "Grad-CAM resolution" is necessary. Does it depend on the layer number or filter/conv. block size? That being said, I find schematic visualization in Fig. 1D quite misleading since the authors use convolution layers rather than dense layers, as shown in Fig. 1. Also, note a mistype "ResNet -> Resent" in the caption. Please, revise.
3. Please, show examples of generated samples from the dataset with different disorder levels. On page 10 (first paragraph), it is also mentioned that "the fixed disorder dataset has been divided according to the fracture strain into three groups of the same size". Please clarify how exactly the fracture strain was determined, and what specific task these three groups were used for?
4. On page 4, the authors write "for each configuration we record ... the final fracture path (for definition of these terms see the Materials and Methods)", however, I could not locate this definition. In what form (data structure) the final path was stored/outputted from NN? Can its evolution be resolved in the time domain?
5. Authors in multiple parts of the manuscript demonstrate that the prediction potential of models is the best for low strain/low disorder. Could you please elaborate on possible reasons for this observation in the revised version of the manuscript and suggest a potential strategy for improving the prediction in the case of high failure strain?
6. Please check affiliation #2.
7. Finally, I encourage the authors to refine the manuscript's readability by adding more details/schemes of the used ML approaches (at least in SM).

Summarizing, this manuscript presents a new approach to visualize the structure-properties relations in disordered solids using DL and Grad-CAM heatmaps that proves to be capable of capturing bond-breaking locations and even full crack path. This approach is definitely of interest for related future studies on more involved 2D or 3D structures, and therefore it is well-suited for Nature Communications. The noticeable drawback of the current manuscript is the unclear description of the employed ML methods that compromise the potential impact of this paper.

Based on scientific merit, this manuscript can be potentially accepted for publication in Nature Communications after major revision.

REVIEWER COMMENTS

Reviewer #1 (Remarks to the Author):

The authors have performed a machine learning study aiming at predicting the failure of two-dimensional silica glasses. Understanding the fracture resistance of disordered materials is an important challenge and the two-dimensional silica glass is a good model system for eventually understanding the behavior of more complex multicomponent glasses in three dimensions. Despite the complex structure of glasses at different length scales, the prospects of using machine learning, both in this and recent related studies, to predict the fracture response of glasses are exciting.

Indeed, the proposed machine learning algorithms in this study are promising, but there several questions and concerns that the authors should address, especially as it relates to the data acquired from molecular dynamics simulations. Therefore, I cannot recommend acceptance of the manuscript in Nature Communications in its current version.

We thank the reviewer for his/her constructive comments and potential interest in the manuscript. We provide below a detailed answer to the remarks.

1. I agree that the control of ring size distribution can impact the mechanical responses of silica glass, but it appears that all the data sets used for the machine learning calculations are on samples that exhibit brittle fracture behavior as shown in Figure 2. As reported in Ref. 27, ductile fracture behavior can also be observed in 2D silica glass. The authors therefore need to test if the method can also be used for a brittle-to-ductile transition. This would be an important test to include in a revised manuscript. In addition, some crystalline phases of 2D silica should also be included in the data sets to improve the generalization of the prediction.

Figure 2 reported stress-strain curves obtained at very low disorder ($s=0.2$). In that regime, the failure behavior is indeed very brittle. We did, however, perform simulations and analyzed data also for larger disorder values, but we did not report the corresponding stress strain curves in the manuscript. For more disordered structures, the stress-strain curves display indeed a more “ductile” behavior, in agreement with Ref. 27 as we now report in the revised Fig. 2. Hence it is indeed possible to use the method in the brittle and in the ductile phase. As shown in the manuscript (Fig. 3EF), predictions are more accurate at low disorder where the behavior is brittle. We thank the referee for pointing out the relation with the brittle-ductile transitions that missed in the previous version of the manuscript. In the revised manuscript, we discuss this point extensively. We included weakly disordered configurations in the analysis, but not a perfect crystal. Note that predicting the failure location of a perfectly crystalline configuration would be futile, since in a simulation with periodic boundary conditions the failure location would be determined by numerical errors only. Furthermore, there is only a single configuration of this type so including it in the data set would be irrelevant.

2. The motivation for using the Grad-CAM method should be explained in more details. What are the pros and cons of this approach?

The Grad-CAM method provides an indication of the most important structural features that contribute to a successful prediction by an otherwise obscure deep learning method. We discuss this point in the revised manuscript.

3. The authors should better explain the reason for using the Watanabe potential in simulating mechanical behavior of 2D silica.

The Watanabe potential describes very well the properties of silica, especially with open boundary conditions as originally shown by Watanabe who compared MD results with ab initio calculations. For this reason, it is a good candidate to describe 2D silica. We have tested also other potentials commonly used to model silica (i.e. Vashista, Tersoff, COMB10). Simulations of the bilayer show (see inclosed panel) that the Watanabe potential yield the most aligned configuration across the two layers. This feature is in agreement with experiments showing that the two planes are perfectly symmetric in silica bilayers. Furthermore, the mechanical properties obtained with the Watanabe potential are in good agreement with

ab initio calculation (See below). We comment on this issue in the revised manuscript.

The mechanical properties of silica glasses seem to be largely overestimated as shown in Fig. 2 when compared to the results of the following references: 1) Gao, Z., Dong, X., Li, N. & Ren, J. Novel Two-Dimensional Silicon Dioxide with in-Plane Negative Poisson's Ratio. Nano Lett. 17, 772-777 (2017); 2) Zhang, J. Phase-dependent mechanical properties of two-dimensional silica films: A molecular dynamics study. Computational Materials Science 142, 7-13 (2018).

We thank the referee for pointing out the apparent disagreement between our results and previous simulations. By a more careful analysis, we realize that the main discrepancy comes the way stresses are estimated. In our paper, we define stress using a bilayer thickness of .43 nm, which is the distance between two Si atoms in the two planes. This thickness value matches closely the results of ab initio calculations in Gao et al 2017 (see Tables S1-S2 in that reference) where the stress was defined using a larger effective thickness of .74 nm, yielding a lower estimate of the Young modulus. If we use the same value of the thickness as Gao et al. our estimates for the Young modulus will be much closer (178 GPa for Gao et al. against 190 GPa in our case). Zhang et al estimate a Young modulus of 416GPa but do not provide the value of the thickness they use, making a quantitative comparison impossible. To avoid the ambiguities associated with defining a thickness for a 2D material, in the revised manuscript we now measure stress and Young modulus in N/m as it is customary in other quasi-2D materials such as graphene. We also compare explicitly our results with ab initio calculations.

4. The relationship between the brittleness and disorder should be provided and discussed.

As discussed above, disordered configurations are less brittle than ordered ones. In the revised Fig. 2, we provide an illustration of the dependence of brittleness on disorder.

5. Analyses of rupture and fracture are based on the breakage of Si-O bonds, but oxygen atoms are invisible in the provided images. The authors should therefore discuss whether – and if so, how – the oxygen atoms play a role in the machine learning fracture prediction.

The oxygen atoms are treated in the point-wise SVM analysis (see Fig. 2B) and are also present in the images analyzed by ResNet although they are not very visible. The image was obtained to make it similar to TEM images where oxygen atoms are barely distinguishable. This issue should not be relevant since our lattices are constructed in a way so that all the Si atoms were 4-coordinated, and therefore adding the O atoms in-between the Si-Si bonds would not add any structural information. In practice, the “topology” of the lattice – that we claim being the relevant

information for failure prediction – is already defined by the sole Si atoms.

6. *It would be interesting if the author could address whether there is any relationship between the position of first rupture and the full fracture path. It may be helpful for illustrating whether these two properties arise from the same structural origins.*

There is a strong correlation between the location of the final crack path and the position of the first broken bond. This correlation is present independently of disorder as shown in the figure. We now provide a supplementary figure illustrating this point.

7. *As mentioned in the manuscript, both sides of the sample have been included in the data set. The authors should demonstrate whether the two sides of the sample can generate the same prediction.*

As discussed above, by construction and in agreement with experiments the two sides are identical. Hence, they must certainly generate the same predictions. We comment about this issue in the revised manuscript.

8. *For the transfer learning approach to be able to tackle large samples, I am wondering whether there is any dependence on the image scale? For instance, does the prediction still work well for an enlarged (zoomed-in) image?*

This issue is addressed in Fig. S2 where we discuss the dependence of prediction with the resolution of the image.

Minor comments / typos:

9. *In Figure 4, it is suggested to use the diagonal line to compute R2 for accuracy comparison.*

We added a line to the figure.

10. *Page 15: "ensamble"*

Corrected

11. *Page 16: the total of splitting of data is not equal to 100%.*

We have corrected the typo. Many thanks to the referee for perusing the text.

Reviewer #2 (Remarks to the Author):

The manuscript "Predicting the failure of two-dimensional silica glasses" addresses a fundamental issue of the failure in disordered solids, attempting to connect peculiarities in 2D structure of silica glass with the observed specific failure strain, location of the first bond break event, and even crack path. Understanding such structure-properties relationships is an involved task in amorphous disordered structures partially due to the absence of classical defects. Because of that, machine learning techniques have become increasingly popular to predict the behavior of disordered solids. The authors address this in the introduction and apply both local (SVM) and global methods (ResNet + Grad-CAM) to study failure in 2D silica glass. In my opinion, the most prominent new result that authors also highlight in the abstract is the employment of Grad-CAM to build attention maps associated with the failure location and crack path. Such Grad-CAM maps provide very good interpretability of the ML predictions and might be

a promising tool for future studies in the related areas. Authors show a correlation between attention maps, disorder parameters, and failure location, although I would rather say these correlations are more of topological nature than physical. These particular results, in my opinion, provide enough novelty to consider publication in Nature Communications.

We thank the referee for his/her positive assessment.

The manuscript relies on a well-established methodology to generate and strain atomic structures for further study, and I think that all necessary details are provided in the "Materials and Methods" part of the manuscript. The methods related to machine learning are also relatively standard (regarding computer science rather than materials science). However, since these methods (SVM, CNNs, and Grad-CAM) are employed in a non-standard domain, they require much better explanation and visualization than provided in the current manuscript. I would like to emphasize that this is a major issue preventing the acceptance of the manuscript in its current form. I have an impression that authors prefer to demonstrate as many results as possible, somewhat sacrificing the clearness and completeness of the presentation.

We have improved the manuscript following the suggestions of the referees.

To improve the manuscript, authors should address the following issues (in no particular order):
1. Please provide an exact architecture of ResNet50 used in this study. On page 19, the authors mention a 4x4 heatmap that corresponds to the last convolutional block and a 32x32 convolution block that corresponds to the first convolution block. It is unclear why these exact dimensions are used without seeing ResNet50 architecture in the manuscript. Moreover, even after analyzing classical ResNet50, I do not understand where these numbers come from. ResNet50 uses 224x224x3 images as input, while in the presented manuscript, the input dimensions are 128x128x1. Please, clarify.

We better describe the architecture of ResNet50 in the revised manuscript including also two additional diagrams. ResNet50 is a very complex deep convolutional neural network architecture, comprising many kinds of layers and skip connections. However, the size of convolutional filters remains constant through a set of layers, named 'convolutional blocks' (the name assigned by the keras library to each layer starts with the relative convolutional block). In the present case of study, the input image has a size of 128x128 pixels. Notice that ResNet50 can handle any linear input size bigger than 32 pixel. First, the image is elaborated and transformed in filters of size 64x64. Then those filters are processed through the convolutional blocks and are transformed in filters of 32x32, 16x16, 8x8 and 4x4 dimensions. Grad-CAM heatmaps can be thought as pixels-wise derivatives of a given convolutional layer with respect to the ResNet final output. In order to understand the decision process made by ResNet50, we have focused on the convolutional layers at the end of each convolutional block. The Grad-CAM heatmap from the first convolutional block is an heatmap of size 32x32, the Grad-CAM heatmap from the second convolutional block is an heatmap of size 16x16 and so on. In order to highlight the important part of the input image (the configuration image of 128x128 dimensions), we have resized the Grad-CAM heatmaps to a size of 128x128 pixels and superimposed them on the input images.

2. As a continuation of issue #1, a more rigorous definition of the term "Grad-CAM resolution" is necessary. Does it depend on the layer number or filter/conv. block size? That being said, I find schematic visualization in Fig. 1D quite misleading since the authors use convolution layers rather than dense layers, as shown in Fig. 1. Also, note a mistype "ResNet -> Resent" in the caption. Please, revise.

We provide a more detailed discussion in the method section of the revised manuscript. The resolution of the Grad-CAM heatmap coincides with its original linear dimension. As explained in the previous point, Grad-CAM originates from the last convolutional layer of a convolutional block, which has a defined linear dimension. For instance, the Grad-CAM heatmap generated from the first convolutional block has a linear dimension of 32 (the first convolutional block contains filters of 32x32 pixels) and so on. The first convolutional block captures coarse-grained

information and therefore Grad-CAM resolution is high. In contrast, the last convolutional block captures very general and macroscopic information yielding a low resolution Grad-CAM heatmap. The schematic visualization in Fig. 1D has been revised to better reflect the actual structure of the algorithm. The typo in the caption has been corrected.

3. Please, show examples of generated samples from the dataset with different disorder levels. On page 10 (first paragraph), it is also mentioned that "the fixed disorder dataset has been divided according to the fracture strain into three groups of the same size". Please clarify how exactly the fracture strain was determined, and what specific task these three groups were used for?

We report now examples at different disorder in the revised Fig. 2. 'Fracture strain' was determined as the strain at which the first bond was broken. In the revised manuscript, we use the term 'rupture strain' in order to avoid confusion.

4. On page 4, the authors write "for each configuration we record ... the final fracture path (for definition of these terms see the Materials and Methods)", however, I could not locate this definition. In what form (data structure) the final path was stored/outputted from NN? Can its evolution be resolved in the time domain?

We now add a more precise definition on how the fracture path is defined. The final path was stored as an image. An example is provided in Fig. S7A. We do not consider the evolution of the crack path, but only the final path after fracture.

5. Authors in multiple parts of the manuscript demonstrate that the prediction potential of models is the best for low strain/low disorder. Could you please elaborate on possible reasons for this observation in the revised version of the manuscript and suggest a potential strategy for improving the prediction in the case of high failure strain?

As discussed in the answer to referee 1, samples at strong disorder/high failure strain fail in a less brittle fashion. This means that the first bond to break often does not correspond to catastrophic failures. These features make predictions harder. We discuss this point in the revised manuscript.

6. Please check affiliation #2.

OK

7. Finally, I encourage the authors to refine the manuscript's readability by adding more details/schemes of the used ML approaches (at least in SM).

We have now added two detailed schemes relating to the artificial neural networks used.

Summarizing, this manuscript presents a new approach to visualize the structure-properties relations in disordered solids using DL and Grad-CAM heatmaps that proves to be capable of capturing bond-breaking locations and even full crack path. This approach is definitely of interest for related future studies on more involved 2D or 3D structures, and therefore it is well-suited for Nature Communications. The noticeable drawback of the current manuscript is the unclear description of the employed ML methods that compromise the potential impact of this paper. Based on scientific merit, this manuscript can be potentially accepted for publication in Nature Communications after major revision.

We hope the revised manuscript is able to overcome the limitations pointed out by the referee.

REVIEWERS' COMMENTS

Reviewer #1 (Remarks to the Author):

I am satisfied with the authors' responses to the concerns raised in the review reports and I therefore believe the current version of the manuscript is suitable for publication.

Reviewer #2 (Remarks to the Author):

The authors of the manuscript "Predicting the failure of two-dimensional silica glasses" have successfully addressed all my comments. The updated figures and new schemes for neural networks make the manuscript more accessible for reading. I fully support the publication of the revised version in Nature Communications.

At the proofreading stage, please capitalize Grad-Cam in the caption for Figure 4I.